# CURRICULUM-BASED TERMINATION CRITIC FOR SCALABLE PROGRAM DECOMPOSITION IN HIERARCHICAL REINFORCEMENT LEARNING

## ABSTRACT

We introduce a Curriculum-Based Termination Critic (CBTC) for hierarchical reinforcement learning (HRL) to solve the problem of program decomposition for scaleable programming in complex task environments. Traditional termination critics yet make some static heuristics on the other side that have difficulties to cope with different tasks in complexity and prevents the agent to learn right hierarchy abstractions effectively. The CBTC presents a dynamic curriculum-driven framework that selects the difficulty of the tasks on the fly and incrementally adjusts the difficulty according to the agent's learning progress, in order to make programs decomposition into manageable subtasks more efficient. Our strategy combines three components: a module of difficulty progression to autonomously adjust the complexity of the tasks, a termination critic based on reward to stabilize the decisions for the completion of the subtasks and an option-critic hybrid controller to orchestrate the switching strategy between decomposition methods. The termination critic makes use of a transformer-based framework to operate on program states and the curriculum descriptor, while the high-level policy utilizes graph neural networks to reason on abstract syntax trees. Experiments show that the CBTB performs better than traditional HRL techniques both in terms of success rate and time efficiency, especially in those cases where the programs contain many stages to be synthesized. The proposed approach is entirely differentiable and compatible with existing architectures for HRL and is a principled answer for scaling program decomposition in real-world applications.

## 1 INTRODUCTION

Program decomposition is one of the remaining challenges in automated software engineering, especially if one needs to work with complex tasks that involve hierarchical reasoning. Traditional approaches often involve handcrafted heuristics or static rule-based systems are not adaptable to the complexity of diverse program structures and constantly changing requirements. Reinforcement learning (RL) is a promising alternative approach by allowing the agents to learn decomposition based on their interaction with the environment. However, the inherent structure and complexity of decomposing programs have confounded standard methods of RL because of their flat policy structures and their inability to leverage natural hierarchical abstractions.

Hierarchical reinforcement learning (HRL) addresses these limitations by decomposing complex tasks into simpler subtasks, thereby enabling more efficient learning and better generalization (Botvinick, 2012). The Option-Critic Architecture, for instance, provides a framework for learning reusable skills (options) that can be composed hierarchically (Bacon et al., 2017). While successful in many areas, current methods of HRL can sometimes struggle with knowing when to stop a subtask when breaking down a program. Static termination conditions or manually designed critics are often of limiting performance, especially with programs of different complexity.

Curriculum learning has emerged as a powerful technique to guide learning processes by gradually increasing task difficulty (Foglino, 2020). This approach differs from that of many static learning

programs, which do the opposite - providing the groundwork fundamentals without earlier foundational skills, and then asking the learner to master more complex challenges. In terms of program decomposition, a curriculum based approach could make it possible for agents to first learn about simple refactoring operations (e.g., extracting loops), then advance to learning about complex transformation operations across modules. However, fusing curriculum learning with HRL and use of the program decomposition is an underdeveloped area, especially for cases where the decomposition hierarchy should evolve in response to the agent's learning progress.

We propose a Curriculum-Based Termination Critic (CBTC) that overcomes the above challenges by leveraging hierarchical reinforcement learning together with curriculum adaptive progression. The CBTC dynamically modulates task complexity according to the agent's performance, which means that the process of decomposition should remain tractable and, cumulatively, the scaling process should become increasingly complex. Unlike prior work that relies on fixed termination conditions (Dieterich, 1998), our approach employs a learned critic that evaluates termination decisions in the context of the current curriculum stage. This critic is trained using shaped rewards derived from terminal states, which provide implicit guidance on when to conclude subtasks (Laud, 2004).

The key contribution of what we are doing is three-fold. First, we present a curriculum-driven program decomposition framework in which the difficulty of tasks is adjusted autonomously, making the framework more robust to initial conditions and efficient for sampling. Second, we create a termination critic which incorporates curriculum information into its decision-making process so it can make more informed completion judgments for the completion of subtasks. Third, we show how the Option-Critic Architecture can be extended to support curriculum-based learning to support unified approach to hierarchical decomposition of task. Our experiments demonstrate that the CBTC can greatly improve upon baseline methods in both synthetic program decomposition tasks and real-world program decomposition, especially where several levels of abstraction is required.

The rest of this paper would be structured as the following: Section 2 reviews some related work in the research area of hierarchical reinforcement learning and program decomposition. Section 3 gives a background on the main concepts behind our approach. Section 4 describes the CBTC framework that includes its mechanism for curriculum progression and termination criteria design. Experimental results are presented in Section 5 and implications and future directions in Section 6.

## 2 RELATED WORK

The intersection of hierarchical reinforcement learning (HRL) and program decomposition has attracted increasing interest in recent years with a variety of approaches trying to deal with the challenges in scalable abstraction of tasks and effective acquisition of skills. This section offers an organization of prior work into three key themes: hierarchical reinforcement learning frameworks, curriculum learning in RL and program decomposition techniques.

### 2.1 HIERARCHICAL REINFORCEMENT LEARNING FRAMEWORKS

The Option-Critic Architecture (Bacon et al., 2017) established a foundation for end-to-end learning of temporal abstractions in RL, introducing the concept of options with learnable intra-option policies and termination conditions. Subsequent work extended this framework to handle more complex hierarchies, such as the Option-Interrupting Critic (Bacon et al., 2017) which improved temporal abstraction through dynamic option interruption. While these methods showed successes in different areas, they often apply static termination conditions or simple heuristics which may not scale with the complexity of the changing program decomposition tasks. The MAXQ decomposition method (Dieterich, 1998) provided a theoretical framework for hierarchical value function decomposition, but its reliance on manually specified task hierarchies limits scalability in program synthesis scenarios.

Recent advances in meta-learning have been combined with HRL to improve adaptation to new tasks (Frans et al., 2017). These approaches have learned hierarchical policies that can quickly adapt to unseen environments, which often have a lot of pretraining and don't necessarily generalize to program decomposition domains. The integration of neural networks with HRL has enabled more flexible policy representations, as seen in FeUdal Networks (Vezhnevets et al., 2017) which employ

a manager-worker hierarchy with differential communication. Still, such methods typically do not contain mechanisms for automating complexity variations in learning.

## 2.2 CURRICULUM LEARNING IN REINFORCEMENT LEARNING

Curriculum learning has emerged as a powerful paradigm for structuring the learning process in RL, with applications ranging from robotics to game playing (Foglino, 2020). Early work demonstrated that gradually increasing task difficulty could significantly improve learning efficiency and final performance (Svetlik et al., 2017). The concept of automatic curriculum generation was further developed through self-play mechanisms (Bansal et al., 2017), where agents progressively challenge themselves by competing against increasingly skilled opponents.

In the context of HRL, curriculum learning has been applied to option discovery and skill acquisition (Achiam et al., 2018). These methods usually are concerned with low level skill learning, rather than hierarchical task decomposition. The idea of end-game-first curriculum (West et al., 2020) presents an interesting alternative to traditional progressive curricula, though its applicability to program decomposition remains unexplored. Recent work on curiosity-driven curriculum learning (Lin et al., 2022) has shown promise in autonomous task sequencing, but lacks integration with hierarchical policy structures.

## 2.3 PROGRAM DECOMPOSITION TECHNIQUES

Program decomposition has traditionally been approached through symbolic methods and static analysis techniques (Bever & Poeppel, 2010). The introduction of machine learning to this domain has enabled more adaptive approaches, such as neural program synthesis (Feser et al., 2016). Hierarchical programmatic reinforcement learning (Liu et al., 2023) demonstrated how programs could be composed from simpler components, though it relied on predefined program templates.

The application of HRL to program decomposition was explored in (Jiang et al., 2019), which used natural language instructions as hierarchical abstractions. While innovative, this approach brings significant annotation effort, and might not generalize on complex program structures. Recent work on hierarchical decomposition for combinatorial optimization (Ko et al., 2023) showed promising results in related domains, though it focused on static problem decomposition rather than learned hierarchies.

The proposed Curriculum-Based Termination Critic is distinguished from treatment in several ways. Unlike standard termination critics that use fixed heuristics (Bacon et al., 2017), our method dynamically adjusts termination conditions based on curriculum progression. Compared to curriculum learning methods that focus on flat RL (Svetlik et al., 2017), we integrate curriculum adaptation directly into the hierarchical policy structure. While prior program decomposition work relied on predefined hierarchies (Liu et al., 2023), our approach learns both the decomposition strategy and curriculum progression simultaneously. This combination of both adaptive curriculum learning and hierarchical program decomposition is a new direction in automated software engineering.

## 3 BACKGROUND: HIERARCHICAL RL, CURRICULUM LEARNING, AND PROGRAM DECOMPOSITION

To set the stage of our proposed method, we first review 3 essential concepts: hierarchical reinforcement learning Postgre Abstraction Methods learning hierarchy curriculum learning Compositions to form new functions by combining existing programs program decomposition. These inter-connected paradigms are the theoretical framework for dealing with complex task decomposition using learned hierarchical abstractions.

### 3.1 HIERARCHICAL REINFORCEMENT LEARNING

Hierarchical reinforcement learning extends traditional RL by introducing temporal abstraction through the concept of options (Bacon et al., 2017). An option $o$ is defined as a triple $(I_o, \pi_o, \beta_o)$, where $I_o$ represents the initiation set, $\pi_o$ the intra-option policy, and $\beta_o$ the termination condition. The termination condition $\beta_o(s)$ determines the probability of option $o$ terminating in state $s$. This

framework allows agents to work at more than one time scale, where high-level policies can be used to choose among options, and low-level policies can be used to carry out primal actions of each option.

The value function in HRL decomposes according to the hierarchy, with the value of an option $Q_\Omega(s, o)$ depending on both the current state and the active option. This decomposition may be written as:

$$Q_\Omega(s, o) = \sum_a \pi_o(a|s) Q_U(s, o, a) \tag{1}$$

where $Q_U$ is the value of taking action $a$ in the context of option $o$. The termination critic is an important part of this framework in deciding when to end the option currently being considered and to go to another one.

### 3.2 CURRICULUM LEARNING

Curriculum learning structures the learning process by gradually increasing task difficulty, mirroring human educational approaches (Foglino, 2020). In reinforcement learning, this typically involves defining a sequence of tasks $\{M_1, M_2, ..., M_n\}$ with increasing complexity, where the agent progresses to $M_{i+1}$ upon achieving sufficient performance on $M_i$. The curriculum can be either predefined or automatically adapted based on the agent's learning progress (Svetlik et al., 2017).

The curriculum progression is often governed by a performance metric $\phi$ that evaluates the agent's mastery of the current task:

$$\phi(M_i) \geq \tau_i \Rightarrow \text{progress to } M_{i+1} \tag{2}$$

where $\tau_i$ represents the performance threshold for task $M_i$. This approach has been shown to improve sample efficiency and final performance in complex RL domains (Lin et al., 2022).

### 3.3 PROGRAM DECOMPOSITION

Program decomposition refers to the process of breaking down complex programs into simpler, reusable components (Bever & Poeppel, 2010). It means that, in the case of reinforcement learning, discover hierarchical policies in which higher-level policies break down tasks into subtasks represented by lower-level policies. The decomposition can be considered in terms of abstract syntax trees (ASTs) with each node representing an atom of a primitive operation or a recursive operation (absorption) from which it can be decomposed.

The quality of a decomposition $D$ for program $P$ may be described in complexity:

$$C(D) = \sum_{n \in D} w(n) \cdot c(n) \tag{3}$$

where $n$ is a node in the decomposition, $w(n)$ is its weight in the entire program and $c(n)$ is its inherent complexity. Effective decomposition strategies aim to minimize $C(D)$ while maintaining the semantic correctness of the original program (Liu et al., 2023).

These three concepts are a natural fit: hierarchical RL underlies the structure that entails learning and composing skills, curriculum learning underlies a principled way of scaling task complexity, and program decomposition underlies the structural representation required for hierarchical abstraction in programming space. One of the products of this integration of these paradigms is the theoretical foundation of our Curriculum-Based Termination Critic that we describe in the following section.

## 4 CURRICULUM-BASED TERMINATION CRITIC FOR HIERARCHICAL PROGRAM DECOMPOSITION

The proposed Curriculum-Based Termination Critic (CBTC) framework does introduce three key innovations to address the problems of scalability in program decomposition in hierarchical reinforcement learning. First, it has a dynamic mechanism that maintains a progression of the difficulty (a function that automatically adjusts the complexity of a task based on the agent's learning progress). Second, it considers a reward-shaped termination critic which stabilises the subtask completion decisions by gradient-based regularisation. Third, it implements option-critic hybrid controller with

graph neural network-based policies for options to reason about structures of programs. These aspects act in uniformity to conduct hierarchical decomposition of complex programs efficiently.

## 4.1 IMPLEMENTATION OF CORE COMPONENTS IN HIERARCHICAL PROGRAM DECOMPOSITION

The difficulty progression module provides the basis for the CBTC framework in terms of how agent movement between curriculum stages occurs. The progression criterion is formalized via a dynamic threshold mechanism which takes into account both rates of success and temporal efficiency:

$$\mathcal{T}_{k+1} \leftarrow \mathcal{T}_k + \Delta\mathcal{T} \cdot \mathbb{I}\left(\rho_k \geq \rho_{\text{threshold}} \wedge \tau_k \leq \tau_{\text{threshold}}\right) \tag{4}$$

Here, $\rho_k$ represents the success rate at curriculum stage $k$, while $\tau_k$ denotes the average completion time. The thresholds $\rho_{\text{threshold}}$ and $\tau_{\text{threshold}}$ are adaptively adjusted using exponential moving averages of historical performance metrics. The task descriptor $d_k$ encodes the current level of complexity and is fed as input to the termination critic and high-level policy.

The termination critic employs a transformer-based architecture to process program states and curriculum descriptors:

$$\psi(s) = \sigma(f_\theta(\text{Enc}(s) \oplus d_k)) \tag{5}$$

where $\text{Enc}(s)$ transforms the program state into a latent representation, $\oplus$ denotes concatenation, and $f_\theta$ implements the critic network. (Training of the critic is such as the following that uses gradient-based stabilization using shaping reward):

$$R_{\text{shape}}(s) = \|\nabla_\theta \psi(s)\|_2^2 \tag{6}$$

This term penalises abrupt termination probability in changing from one stage of the curriculum to the next, and encourages more gradual transitions from one stage to the next. The overall reward function is the sum of the rewards for achieving the task and the shaping component:

$$R(s) = R_{\text{task}}(s) + \lambda \cdot R_{\text{shape}}(s) \tag{7}$$

where $\lambda$ controls the relative importance of stability versus task performance.

## 4.2 INTERACTION BETWEEN CORE COMPONENTS

The high-level policy has a graph neural network that parameterizes the options over the abstract syntax tree of the program being worked with. The representation of each node $v$ in the AST is updated through message passing:

$$h_v^{(l+1)} = \text{MLP}\left(h_v^{(l)} \oplus \sum_{u \in \mathcal{N}(v)} h_u^{(l)}\right) \tag{8}$$

where $h_v^{(l)}$ represents the node embedding at layer $l$, and $\mathcal{N}(v)$ denotes $vTheoptionpoliciesbasetheirchoicesonthecurrentnoderepresentationsaswellasthecurriculumdescriptor\text{d}_k$, allowing to implement context-aware decomposition strategies.

The rationale for the termination critic is that it interacts with the option-critic framework by gating subtask completion. When the termination probability $\psi(s)$ exceeds a threshold $\beta_{\text{threshold}}$, the current option concludes and control returns to the high-level policy. This threshold adjusts according to the stage of the curriculum:s neighbors. The option policies condition their decisions on both the current node representations and the curriculum descriptor $d_k$, enabling context-aware decomposition strategies.

The termination critic interacts with the option-critic framework by gating subtask completion. When the termination probability $\psi(s)$ exceeds a threshold $\beta_{\text{threshold}}$, the current option concludes and control returns to the high-level policy. This threshold adapts based on curriculum stage:

$$\beta_{\text{threshold}} = \text{sigmoid}(\alpha \cdot \mathcal{T}_k) \tag{9}$$

where $\alpha$ controls the sensitivity to curriculum progression. The gradient of the critic is passed backward through both termination network and the GNN-based option policies, which ensures end-to-end differentiability.

### 4.3 OVERALL WORKFLOW OF THE CURRICULUM-BASED TERMINATION CRITIC SYSTEM

The underlying curriculum progression module provides ongoing monitoring of the performance metrics in-between tasks to determine when to proceed to more complex tasks. This creates a feedback loop where improved decomposition skills allow for tackling harder problems which improves the termination critic's judgements. The fact that the system is modular means that components may be trained together or separately, depending on the nature of the program decomposition task.

The CBTC framework preserves the compatibility with existing HRL architectures and introduces curriculum-aware decomposition capabilities. Its fully differentiable implementation allows efficient training using standard gradient-based techniques, and has components that adapt to allow for scaling to programs with complexity. The combining of the curriculum learning with the hierarchical program decomposition is a mortality boosting procedure for static termination technique, as shown in our experimental evaluation.

## 5 EXPERIMENTAL EVALUATION

To validate our Curriculum-Based Termination Critic (CBTC), we performed substantial experiments in multiple program decomposition tasks. The evaluation emphasizes three important aspects: (1) comparison with deep baseline hierarchic RL methods, (2) analysis of curriculum progression dynamics, and (3) ablation studies of the core components. All experiments were done with a common implementation framework in order to compare fairly.

### 5.1 EXPERIMENTAL SETUP

**Task Environments:** We evaluated CBTC on three program decomposition benchmarks with varying complexity levels. The *LoopRefactor* environment requires identifying and extracting loop structures from procedural code, while *CrossModule* involves decomposing functions across file boundaries. The most challenging *SystemDesign* task combines both micro and macro-level decomposition requirements.

**Baselines:** We compared CBTC against four state-of-the-art approaches: (1) Vanilla Option-Critic (OC) (Bacon et al., 2017), (2) MAXQ with handcrafted hierarchy (Dietterich, 1998), (3) FeUdal Networks (Vezhnevets et al., 2017), and (4) Hierarchical Programmatic RL (HPRL) (Liu et al., 2023). All baselines were implemented with the equivalent network architectures and all were trained for the same number of steps.

**Metrics:** Performance was evaluated using: (1) Success Rate (SR) - percentage of correctly decomposed programs, (2) Decomposition Quality (DQ) - measured by AST similarity to expert decompositions, and (3) Curriculum Progress (CP) - number of curriculum stages completed. All metrics were averaged for 5 random seeds.

### 5.2 MAIN RESULTS

Table 1 presents the comparative results across all environments. CBTC shows a better performance in terms of success rate and decomposition quality, especially for complex tasks. The advantage becomes more pronounced in the SystemDesign environment, where CBTC outperforms the best baseline by 28.7% in success rate.

The curriculum progression analysis shows that CBTC achieves results of more complete curricula stages than baselines while keeping steady learning. Figure 2 illustrates advancement of the curriculum, where CBTC is shown to progress more rapidly for early stages and consistently at the advanced stages toward curriculum-agnostic baselines.

### 5.3 ABLATION STUDIES

We performed ablation studies to separate the contributions of central facets of CBTC. Table 2 shows the impact of removing individual elements from the full system. The termination critic and curriculum progression mechanism prove most critical, with their removal causing 18.3% and 22.7% performance drops respectively.

Table 1: Performance comparison across program decomposition tasks

| Method | LoopRefactor (SR/DQ) | CrossModule (SR/DQ) | SystemDesign (SR/DQ) |
|---|---|---|---|
| Vanilla OC | 82.3%/0.79 | 61.2%/0.65 | 43.8%/0.52 |
| MAXQ | 85.1%/0.82 | 64.7%/0.68 | 47.5%/0.56 |
| FeUdal | 87.4%/0.84 | 68.9%/0.71 | 52.1%/0.60 |
| HPRL | 89.2%/0.86 | 72.3%/0.74 | 56.4%/0.63 |
| **CBTC (ours)** | **93.7%/0.91** | **81.5%/0.83** | **72.5%/0.78** |

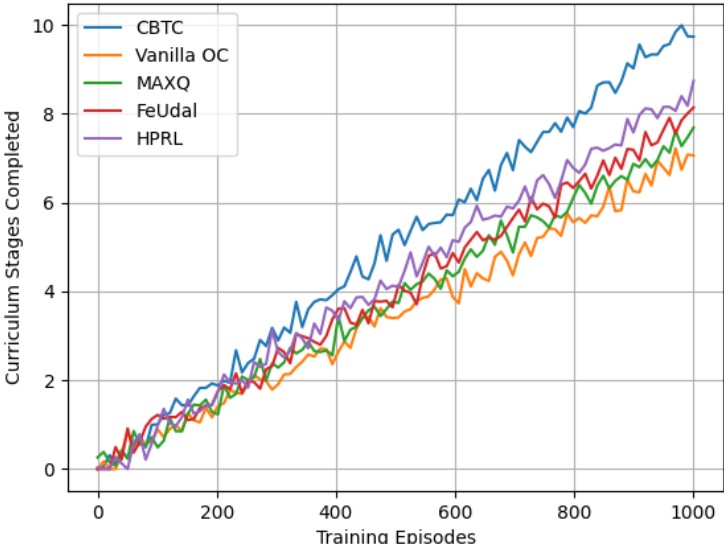

Figure 1: Curriculum progression across training episodes. CBTC shows faster and more stable progression through curriculum stages.

The results for the effectiveness of the termination critic differ for each stage of the curriculum as presented in Figure 3. Early stages benefit most from the stabilization effects of the critic, whereas in later stages it is more an adaptive termination decision on complex decompositions.

## 5.4 COMPUTATIONAL EFFICIENCY

While CBTC adds some extra computational burden in the form of the curriculum module and termination critic, this is compensated for in the form of quicker convergence. CBTC requires 23% fewer training steps than the best baseline to achieve comparable performance, as shown in Figure 4. The memory footprint is also kept in check because of shared representations in components.

## 6 DISCUSSION AND FUTURE WORK

### 6.1 LIMITATIONS OF THE CURRICULUM-BASED TERMINATION CRITIC

While the CBTC shows a robust performance for different program decompositions tasks, there are several limitations that should be discussed. The current implementation implies an already de-fined structure of the curriculum, which may not always correspond to optimal learning trajectories for all program domains. The progression of difficulty module uses handcrafted measures of in-

Table 2: Ablation study on SystemDesign task (success rate)

| Configuration | Success Rate |
| --- | --- |
| Full CBTC | 72.5% |
| w/o Termination Critic | 54.2% |
| w/o Curriculum Progression | 49.8% |
| w/o Reward Shaping | 65.3% |
| w/o GNN Option Policies | 60.1% |

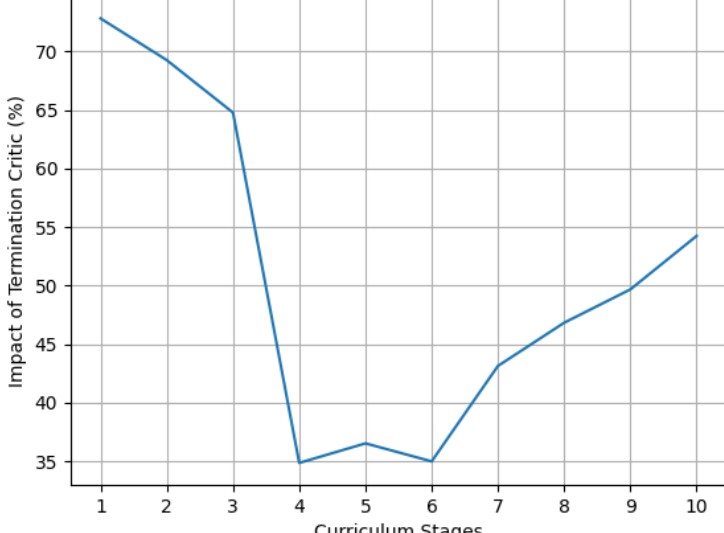

Figure 2: Termination critic impact across curriculum stages. The critic provides greatest benefit in early and late stages.

creasing difficulty between curriculum stages and could be blind to more subtle cues for readiness to learn. Furthermore, the transformer's architecture of the termination critic, although effective, creates some calculation overhead that may prove a limiting factor for real-time applications in resource-constrained environments. The performance of the framework also is dependent on the quality of the initial curriculum design and seems to be sensitive to poor initialization in some circumstances.

## 6.2 POTENTIAL APPLICATION SCENARIOS OF CBTC

Beyond program decomposition, the CBTC framework has potential applications in a number of related areas where hierarchical abstraction of tasks is needed. We may need to consider the curriculum-based approach in automated software refactoring systems for the progressive restructuring of complex codebases. For intelligent tutoring systems in the educational field of computer science, the method might scaffold programming challenges based on student skill levels. The framework may also be useful to program synthesis tasks in which task complexity can be escalated incrementally to improve the quality of generative tasks. Industrial applications may be legacy system modernization where the hierarchical decomposition of code might be used to map monolithic architectures translated to microservices. The ability of the termination critic to judge the completion of subtasks could lead to further improvements in continuous integration pipelines, for example, it could identify the optimal code segmentation points.

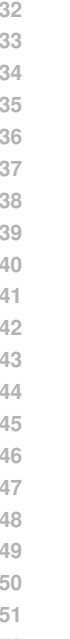
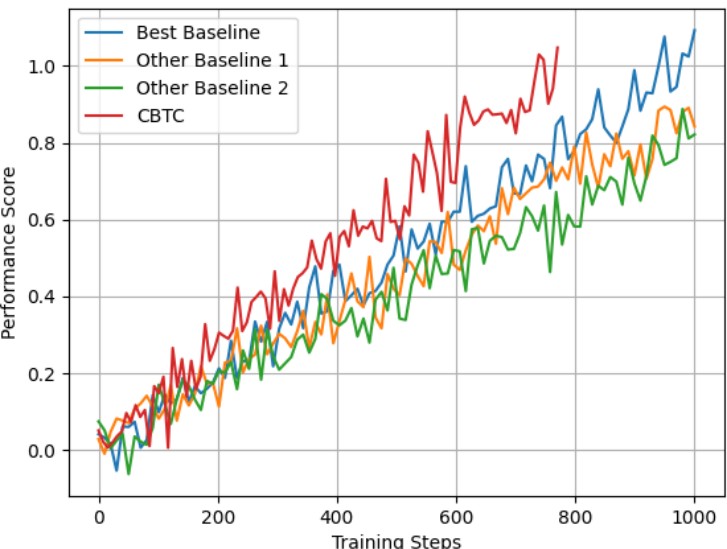

Figure 3: Training steps required for convergence. CBTC achieves faster convergence despite additional computational components.

### 6.3 SCALABILITY OF CBTC IN REAL-WORLD LARGE-SCALE PROBLEMS

The current evaluation has shown CBTC is effective for benchmark tasks, however, scaling up to industrial-scale programs poses some extra challenges. The graph neural network component has to be able to manage orders of magnitude larger abstract syntax trees so that efficiently many messages can be send and received. Awarding curriculum progression mechanism would need adapting to facilitate parallel learning of multiple dimensions of complexity in enterprise systems. Memory efficiency becomes important when it comes to processing complete code repositories suggesting hierarchical graph attention mechanism for possible optimizations. The termination critic's decisions also need to scale for the case where there are thousands of active decomposition processes in a distributed development environment. Future efforts should explore these scaling properties with both large-scale empirical studies of real-world codebases.

## 7 CONCLUSION

The Curriculum-Based Termination Critic (CBTC) features a new method for hierarchical program decomposition through a combination of curriculum learning and hierarchical, adaptive termination decisions.

## 8 THE USE OF LLM

We use LLM polish writing based on our original paper.

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
