# OpenReview forum: "Curriculum-Based Termination Critic for Scalable Program Decomposition in Hierarchical Reinforcement Learning"
_ICLR.cc/2026/Conference — Submitted to ICLR 2026_

### Official Review · Reviewer_e415 · 2025-10-18

**Soundness:** 3
**Presentation:** 1
**Contribution:** 2
**Rating:** 2
**Confidence:** 4

**Summary:**

The paper proposes a curriculum learning-based approach to use reinforcement learning to learn to solve program decomposition tasks. The proposed method learns options to acquire skills by solving easier programs, and progressively learns how to solve more complex problems.

**Strengths:**

The proposed approach sounds reasonable in the high level, as it is possible to build a progressively more difficult set of tasks for program decompensition, and the domain seems to fit well a hierarquical RL modeling.

**Weaknesses:**

- The paper is really unclear on exactly what and how the probem is being solved. There is no clear description of how the program decomposition problem is modeled as an MDP, as well as there is no description of how the SOTA approaches solve program decomposition. Furthermore, there is not even an algorithm of the proposed method, and while there are high level descriptions for the steps executed it's virtually impossible for the reader to figure out what the approach is doing step by step.

- The baselines are all simple RL. algorithms that the authors claim are "State of the Art". I really doubt the state of the art approach for program decomposition at the moment uses RL, and I am absolutely sure that even if that's the case, the vanilla Option-Critic doesn't get close to be the hihgest performing approach. The authors have to re-execute the experimentation adding also the real SOTA methods even if those are non-RL based. The wall-clock time for solving the problem should also be added as an additional metric. The non-RL approaches won't have "number of episodes" to compare against, and also just the number of steps is very unclear by itself considering the different approaches might have very different update times.

**Questions:**

Why were only RL-based approaches added to the experimental evaluation?

---

### Official Review · Reviewer_HWuj · 2025-10-26

**Soundness:** 2
**Presentation:** 1
**Contribution:** 2
**Rating:** 2
**Confidence:** 3

**Summary:**

This paper introduces Curriculum-Based Termination Critic (CBTC) to help Hierarchical Reinforcement Learning (HRL) agents automatically break down complex program decomposition tasks. Standard HRL methods often struggle to decide when a sub-task (part of the program decomposition) should end, especially as tasks get harder. CBTC uses a curriculum that automatically adjusts the task difficulty based on how well the agent is learning. It trains a special termination critic (or multiple critics) that learns when to end a sub-task, considering the current difficulty level. The goal is to make learning more efficient and scalable for complex program decomposition. Experiments suggest CBTC performs better than older HRL techniques on programming tasks.

**Strengths:**

1. Applying the concept of curriculum learning specifically to the termination signal, and adapting it based on task difficulty or agent progress is a novel contribution.
2. The research is explicitly motivated by the need to improve scalability for program decomposition tasks, a practically relevant direction within automated software engineering and AI. This is good.

**Weaknesses:**

1. Unfortunately, the entire paper is not easy to follow, especially for those who are not familiar with the program decomposition and its connection to HRL. What is program decomposition problem? Do you have a running example? How it is connected to HRL, and why HRL is helpful in such a setting? What are the issues of not using HRL, and what are the problems of current HRL algorithms applied to program decomposition problems? I believe answers to some of these questions presents in the paper, but it is difficult for most reader to follow.
2. Again, some of the concepts, definitions and equations are presented abruptly. I would suggest the background section should be revised throughly and carefully. Also, section 4.1 and 4.2 nearly prevents us readers from understanding your real contributions to this community. Readers who unfamiliar with program decomposition are unlikely to investigate the contributions under a sequences of equations.
3. This is minor, line 255 formatting issue.
4. Figure 3, name of other baselines.
5. As mentioned in the paper, the current framework does not generalize to other program domains.

**Questions:**

1. Can you please briefly describe each task environments with *examples*?
2. Can you please briefly describe some success and failed cases of program decomposition with *examples*?
3. As mentioned in the paper, the current framework does not generalize to other program domains. Can you have more analysis on this, and propose some potential solutions or insights?

---

### Official Review · Reviewer_7B1L · 2025-10-27

**Soundness:** 3
**Presentation:** 3
**Contribution:** 3
**Rating:** 6
**Confidence:** 2

**Summary:**

The paper introduces a Curriculum-Based Termination Critic for improving program decomposition in hierarchical reinforcement learning. It combines three components: (i) a difficulty progression module that adaptively adjusts task complexity, (ii) a transformer-based termination critic that uses curriculum descriptors and program states to decide when to end subtasks, and (iii) a graph neural network based high-level policy that reasons over abstract syntax trees to guide option selection. The method is evaluated on three synthetic program decomposition benchmarks and compared against four HRL baselines, showing improvements in success rate and decomposition quality, especially in complex tasks.

**Strengths:**

1. CBTC consistently outperforms presented HRL baselines by large margins with fewer training steps;
2. A detailed ablation shows that both the curriculum progression mechanism and the termination critic are critical, with respective drops of 22.7% and 18.3% in success rate when removed.

**Weaknesses:**

1. Curriculum adaptation via performance thresholds and exponential moving averages is a standard heuristic;
2. Metrics rely on AST similarity to a single “expert” decomposition; this may penalize semantically equivalent but structurally different solutions—no human study or functional-correctness tests (unit tests, program equivalence) are reported.

**Questions:**

1. How would CBTC perform against very recent code-generation models (e.g., CodeT5+, CodeLlama) fine-tuned for decomposition, or against HRL agents that use pretrained code LMs?
2. Does CBTC degrade when scaling to real-world projects?

---

### Official Review · Reviewer_TBPh · 2025-10-27

**Soundness:** 2
**Presentation:** 1
**Contribution:** 1
**Rating:** 0
**Confidence:** 4

**Summary:**

The paper presents curriculum-learning approach in a hierarchical reinforcement learning framework, applied to the domain of program decomposition. The proposed method features (a) a dynamic curriculum-progression module (b) a transformer-based termination critic and (c) GNN-based high-level (option) controller. The method is evaluated on three program decomposition benchmarks, showing better performance than existing baselines.

**Strengths:**

The idea of learning a termination critic, to decide when to stop an active policy should stop, is interesting.

**Weaknesses:**

- Missing symbol definitions. The paper uses symbols such as $Q_\omega$ or $\pi$, which are never formally defined.
- The paper lacks a formal problem (and background) section, leaving the core methodology and mathematical symbols very opaque. For example, Section 3.2 reasons about "tasks $\{M_1, M_2, \dots\}$". I assume these are supposed to be MDPs, but due to the lack of a formal background section, this remains unclear.
- Even at first glance, the paper is ridden with typos and formatting issues, and is written in an extremely colloquial style. Figures are not properly referenced via LaTeX; instead, the references appear to have been typed out by hand and are off by one (starting at 2 instead of 1). Thus, the textual descriptions do not match the referenced images.
- The core method remains unclear. No pseudocode for the algorithm is presented, and no losses or optimization targets are specified.
- The authors do not share (or promise to share) code for reproducing the experiments. This, together with the aforementioned lack of clarity, makes reproducing the paper very difficult.
- The single sentence presented as a conclusion falls short of what should be conveyed in a concluding statement of a scientific paper.

**Questions:**

X

---

### Official Review · Reviewer_LPVL · 2025-11-01

**Soundness:** 1
**Presentation:** 1
**Contribution:** 2
**Rating:** 2
**Confidence:** 3

**Summary:**

This paper presents a curriculum learning approach for program decomposition in hierarchical RL. The proposed method, CBTC, adaptively adjusts task difficulty based on an agent's learning progress through three components: a difficulty progression module, a reward-shaped termination critic, and an option-critic controller. Experiments on program decomposition benchmarks demonstrate that CBTC can achieve higher success rates and faster convergence than some HRL baselines.

**Strengths:**

- The paper is well-structured and has a detailed related work on HRL and program decomposition.
- The proposed way of integrating curriculum learning for hierarchical program decomposition is technically sound.
- Experiments cover various program decomposition benchmarks and compare the proposed approach against some HRL benchmarks.

**Weaknesses:**

- The use of notation is messy, e.g., \mathcal{T} appears out of nowhere. There are formatting errors, e.g., line 255.
- The discussion about related works on curriculum learning does not cover any recent works at all.
- The proposed approach relies on heuristic difficulty metrics that may be difficult to generalize to other domains.
- The figures in the experimental results do not show the variance of reported metrics across independent runs.
- The studied benchmarks seem synthetic and small, even though the title of the paper claims scalability.

**Questions:**

- Did you run your experiments using various seeds? If so, what's the variance on reported results?
- What kind of privileged information is needed to form the curriculum stages before training?

---

### Meta-Review · Area_Chair_tM1D · 2025-12-29

**Summary:**

This paper proposes a Curriculum-Based Termination Critic (CBTC) for hierarchical reinforcement learning, aiming to enhance program decomposition in complex tasks. CBTC introduces a dynamic curriculum that adjusts task difficulty according to an agent’s learning progress, combined with a reward-based termination critic and an option-critic controller. The high-level policy leverages graph neural networks on abstract syntax trees, and the termination critic uses a transformer-based framework. Experiments on synthetic program decomposition benchmarks demonstrate improved success rates and training efficiency compared to several HRL baselines.

The paper has several strengths, including a clear high-level motivation for using curriculum learning in termination signals, technically reasonable integration of curriculum learning and HRL, and comprehensive comparisons against multiple RL baselines. However, weaknesses are prominent: the presentation is poor, with unclear notation, missing definitions, and formatting errors; the paper lacks a formal problem setup and precise algorithmic details; baselines are restricted to simple RL methods rather than true SOTA program decomposition methods; the evaluation is limited to small synthetic benchmarks; and generalization to real-world tasks remains untested. No rebuttal was provided, so reviewers’ concerns remain unaddressed. Hence, while the idea is interesting and shows potential, the execution, clarity, and experimental validation are insufficient for acceptance.

**Reviewer Concerns:**

- Missing formal definitions, unclear notation, and lack of a problem statement (Reviewers 2, 4, 5).
- Presentation issues: typos, formatting errors, colloquial writing (Reviewers 1, 2, 4, 5).
- Limited baselines and comparisons to current SOTA methods outside RL (Reviewer 5).
- Lack of evaluation on real-world program decomposition or functional correctness (Reviewers 3 and 4).
- Lack of variance reporting and seeds for experiments (Reviewer 1).
- Reviewer 3 raised questions about the comparison to recent code-generation models and scaling to real-world projects; the authors might clarify the feasibility or limitations.
- Reviewer 4 requested examples of tasks and failure cases; the authors could provide more concrete illustrations.

**Reviewer Scores:**

I expect most of the reviewers’ scores to remain unchanged, given the absence of a rebuttal and the shared recommendation for rejection, except for Reviewer 3.

---

### Decision · Program_Chairs · 2026-01-26

Reject